# Latent learning: episodic memory complements parametric learning by enabling flexible reuse of experiences

**Andrew Kyle Lampinen**                                        *lampinen@google.com*
*Google DeepMind*

**Martin Engelcke**                                             *engelcke@google.com*
*Google DeepMind*

**Yuxuan Li**                                                   *liyuxuan@google.com*
*Google DeepMind*

**Arslan Chaudhry**                                             *arslanch@google.com*
*Google DeepMind*

**James L. McClelland**                                         *jlmcc@google.com*
*Google DeepMind*
*Department of Psychology, Stanford University*

**Reviewed on OpenReview:** *https://openreview.net/forum?id=RuWGeX5ZiB*

## Abstract

When do machine learning systems fail to generalize, and what mechanisms could improve their generalization? Here, we draw inspiration from cognitive science to argue that one weakness of parametric machine learning systems is their failure to exhibit *latent learning*—learning information that is not relevant to the task at hand, but that might be useful in a future task. Using controlled, synthetic benchmarks, we show how this perspective links failures ranging from the reversal curse in language modeling to new findings on agent-based navigation. We then highlight how cognitive science points to episodic memory as a potential part of the solution to these issues. Correspondingly, we show that a system with an oracle retrieval mechanism can use learning experiences more flexibly to generalize better across many of these challenges—thus motivating episodic memory as an important direction for research in AI. We also identify some of the essential components for effectively using retrieval, including the importance of *within-experience* in-context learning for acquiring the ability to use information *across* retrieved experiences. In summary, our results illustrate one possible contributor to the relative data inefficiency of current machine learning systems compared to natural intelligence, and help to understand how retrieval methods might complement parametric learning to improve generalization. We close by discussing some of the links between our work and findings in cognitive science and neuroscience—including a possible perspective on hippocampal contributions to generalization—and the broader implications.

## 1 Introduction

When do AI systems fail to generalize like natural intelligence? This question has been central to AI, and has only redoubled with the successes of deep learning over the last decade (e.g., Lake et al., 2017). For example, language models (LMs) show surprising failures to generalize to reversals of their training, e.g., failing to generalize from "Plato taught Aristotle" in training to "Aristotle's teacher was Plato' at test (Berglund et al., 2024)—though the models can readily make these generalizations in context (Lampinen et al., 2025).

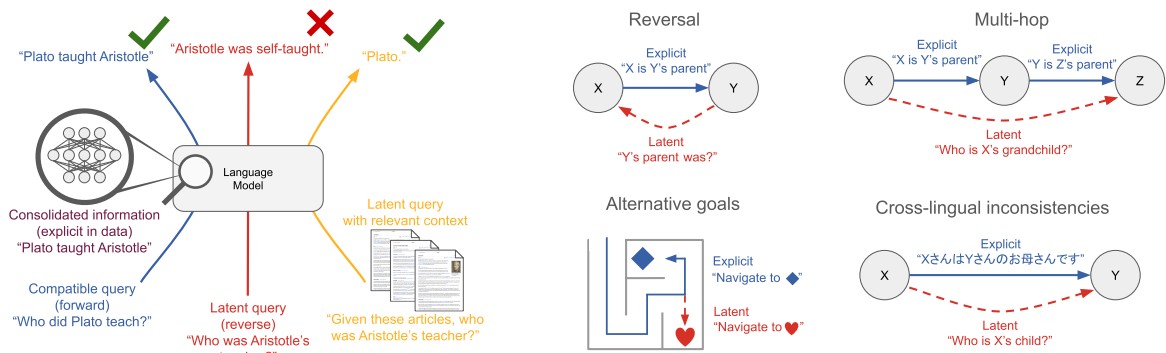

(a) The reversal curse.

(b) Examples of explicit and latent information or tasks.

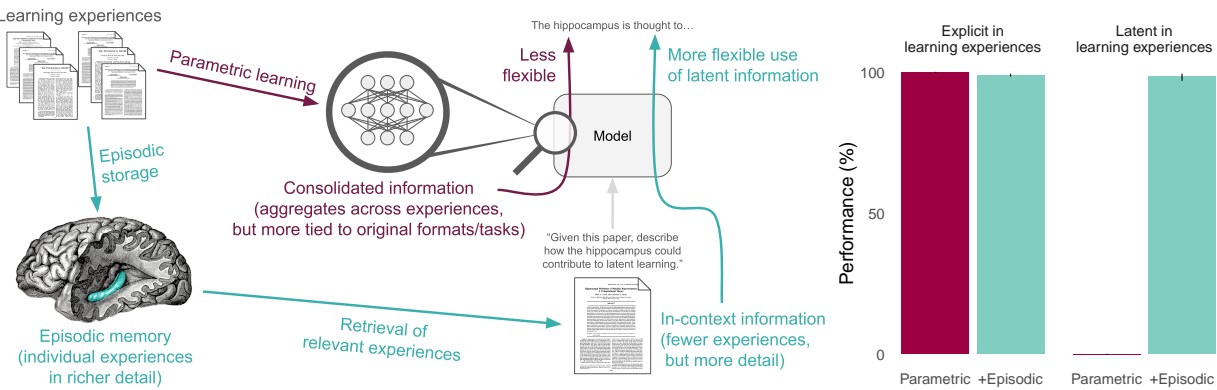

(c) Episodic memory complements parametric consolidated knowledge.

(d) Resulting performance.

Figure 1: Conceptual overview of the challenges of using latent information from training experiences, and how retrieval complements parametric learning to overcome them. (a) The reversal curse (Berglund et al., 2024) is an example of how parametric learners, such as language models, consolidate information in ways that depend on the learning task and format. Models that learn a relation in one format can answer queries compatible with the learning format, but not those that reverse the relation—even though the reversed relation is latently implied by the forward one, and the models are fully capable of reversing relations to make inferences in context (Lampinen et al., 2025). (b) Challenges of reversal are one instance of the much broader phenomenon that what is explicitly learned may also latently convey information relevant to other tasks—e.g., multi-hop reasoning, alternative goals, or answering questions in other languages. Like the reversal curse, learning on such sequences may primarily improve performance on the explicit information or goals; however, if the sequence were in context, models would readily be able to make inferences about the latent information. (c) Therefore, explicit retrieval of specific experiences from episodic memory complements the broader knowledge of parametric learning—by making select, relevant experiences available in context where the latent information they contain can be used more flexibly, in ways different from the original task setting in which they were encountered. (d) Thus, we will typically expect systems with solely parametric learning to perform well at new tests of knowledge that is explicit in learning experiences, but we expect *selective* performance advantages for episodic retrieval in tests of knowledge or tasks that are latent in learning experiences—as we demonstrate below. (These illustrative results are adapted from the simple reversals experiments below.)

Why do models fail to make these generalizations outside the current context, and how could this failure be addressed?

Here, we draw on work in cognitive science to identify a key theme that we believe unifies and generalizes some of these past findings, and points towards a solution. Specifically, we identify the capacity for *latent learning* (Blodgett, 1929; Tolman, 1948) as a key gap between natural and artificial intelligence. Latent learning is the ability of a system to learn information that is not relevant to the task at hand, but that might be useful for a different future task—which can also be seen as a way of implementing a form of "prospective learning" (De Silva et al., 2023). We suggest that AI systems fail to exhibit latent learning in most cases—that is, AI only learns information insofar as it is relevant to the *current* task, and only *applies* information it has learned in the past insofar as it is *explicitly cued* by the current task or available through *similarity-based* or *process-based* generalization from it. This is *not* to say that parametric learning does not generalize well in many cases; one of the key features of parametric learning is its ability to integrate learning over many experiences to improve generalization. Indeed, there is a long history of connectionist research documenting the generalization capacities of parametric learning via semantic similarity (e.g. Hinton, 1986; Rogers & McClelland, 2004) or learning generalizable procedures or functions, (e.g Rumelhart & McClelland, 1986). However, we will argue that this type of task-driven parametric learning is distinct from the latent learning of *specific information* from prior experiences that could be useful in a *sufficiently different* task at a later time. We elaborate this distinction in the Background section below.

As a concrete example of a latent learning challenge, consider again a language model that in training learns that "Plato taught Aristotle," but does not encounter any other facts about Plato or Aristotle. The model will likely be able to answer questions like "Who did Plato teach?" correctly with "Aristotle." Surprisingly, however, it will not be able to answer "Who was Aristotle's teacher?" (Berglund et al., 2024). Crucially though, if the forward information is available in context (e.g., by placing the training document containing the statement "Plato taught Aristotle" in the model's context), the model will be able to answer the reverse question "Who was Aristotle's teacher?" (Lampinen et al., 2025) in context. Thus, the model in some sense knows the pieces it needs to solve the reversal: it can state the forward direction or answer questions about it, and it has learned a generalizable procedure for answering reversal questions when the forward direction is available in context. However, the way that it encodes the forward direction in its parameters does not allow it to generalize to the reversal task *without* the forward information in context. We see this as being an instance where the forward knowledge is *explicit* in the data, whereas the reversal is *latent*—the explicitly trained information is encoded in the model's parameters, and available in other contexts, but the model cannot use the parametric information flexibly to access what is latent in it. However, the greater flexibility of in-context learning enables accessing the latent information. We will provide a more formal outline of this perspective below, and link it to a broader set of phenomena including logical inferences and generalization to novel navigation goals—unifying a number of previous findings about successes and failures of generalization and highlighting new ones.

Given this argument, we again take inspiration from cognitive science to suggest a path to a solution. Specifically, in natural intelligence, episodic memory and parametric learning are complementary (McClelland et al., 1995; Kumaran et al., 2016). Episodic memory seems to play an important role in some types of latent learning (Kimble & Greene, 1968; Myers et al., 2000) and other types of generalization (Bayley & Squire, 2002; Eichenbaum & Fortin, 2009). Moreover, in some of the cases where AI fails to generalize from its parametric learning, it generalizes well in context (Lampinen et al., 2025), suggesting that a system that uses episodic memory to reinstate the relevant information into context could generalize better. Motivated by this intuition, we test a model endowed with an oracle episodic retrieval system during both training and testing, and show that it improves on many of the latent learning failures of current methods. Importantly, generalization from episodic retrieval does not require estimating at encoding time which information will be useful or in what way; instead, storing veridical episodes allows more flexible access at test time than can be obtained through parametric generalization alone. While at some level, this point is just a restatement of the fact that Retrieval Augmented Generation (RAG; Lewis et al., 2020) can be useful, we believe that seeing it in the context of latent learning provides a new perspective on *why* retrieval is useful, and how current AI systems differ from natural intelligence. It therefore points to an important direction of research for AI:

building more effective episodic memory and retrieval systems that approximate the role of the hippocampus in enabling generalization.

In summary, this paper makes the following contributions:

- Identifying *latent learning* of information that may be useful in future tasks as a key gap between natural and artificial intelligence.

- Demonstrating this gap through controlled synthetic benchmarks, and reinterpretations of prior results.

- Arguing that episodic memory (or nonparametric retrieval) can bridge this gap by making relevant information available to more flexible online systems.

- Confirming empirically that oracle retrieval can help to overcome this gap, thus motivating episodic memory systems as an important area of research for AI.

- Highlighting features that contribute to learning to use retrieved information—particularly the importance of *within-episode* in-context learning experience for learning to use information *across* retrieved episodes.

- Relating these findings to the potential role of offline and online episodic replay in enabling certain types of flexible generalization in natural intelligence.

These results shed light on some differences between natural and artificial intelligence, suggest new perspectives on extant practices such as RAG, and may inspire new research in cognitive science and AI.

## 2    Background

We first review some of the background and related research, and attempt to situate our work within it. To this end, we begin with the history of latent learning and complementary learning systems in cognitive science and neuroscience. We then turn to perspectives on (meta-)learning and generalization in artificial intelligence, with a particular focus on generalization within and across contexts in transformer language models. Finally, we discuss the difference between the well-established types of similarity-based and procedural generalization in connectionist and modern models, and the latent learning challenges we discuss here.

### 2.1    Latent learning & episodic memory in natural intelligence

**Latent Learning:**    The concept of "latent learning" was introduced by Blodgett (1929), and subsequently elaborated by Tolman (1948), to describe particular phenomena exhibited by rats exploring mazes: the rats seemed to *latently* learn information that was not relevant to their current goals, but that might be useful in the future. For example, if rats were neither hungry nor thirsty, they would ignore any water or food encountered in the maze. However, if they were later placed in the maze when they *were* hungry or thirsty, they would rush to the appropriate locations. Thus, even though the rats had *not* been motivated to seek either food or water when initially exploring the maze, they had *latently learned* the locations of those resources in a way that allowed them to efficiently exploit them in the future.

Latent learning has been suggested to depend in part on the medial temporal lobe (MTL; including the hippocampus) and lesions to this region impair latent learning in some cases (Kimble & Greene, 1968; Myers et al., 2003), as well as other types of generalization, such as transitive inferences (Eichenbaum & Fortin, 2009) or linguistic or semantic generalizations (Bayley & Squire, 2002). (Note that some latent learning processes may not be MTL-dependent; Kimble et al., 1982; Myers et al., 2000.) More recent work has highlighted that hippocampal "preplay" might help to cache the solutions to potential future tasks (Ólafsdóttir et al., 2015; Carvalho et al., 2025). Overall, these results suggest that episodic memory may contribute to some—if not all—forms of latent learning in natural intelligence.

**Complementary learning systems:**    One strand of hippocampal research focuses on "complementary learning systems" (McClelland et al., 1995; Kumaran et al., 2016)—the idea that the rapid learning and episodic memory of the hippocampus provides benefits that complement the slower, more generalized learning

within the neocortex. The original argument for why the rapid-learning system is useful was that it could rapidly acquire new information without interfering with broader knowledge in the slower system; these pieces of rapidly-acquired information could then be recalled when needed for immediate task performance, and integrated into the slower-learning system over time via recall or replay—which naturally results in interleaving the learning with other experiences and thereby reduce catastrophic interference.

Our work suggests a potential complementary benefit of episodic memory: it could support more flexible use of past experiences in contrast to cortical learning, while cortical learning might be more closely tied to some aspects of the task or format in which the original experiences occurred. This suggestion is consistent with the lesion findings above, and more broadly compatible with arguments that the hippocampus plays a crucial role in organizing (spatial) knowledge into structures that support flexible learning and behavior (Behrens et al., 2018; Raju et al., 2024; Sun et al., 2023)—and in linking memories together to enable generalizations such as transitive inferences (Eichenbaum & Fortin, 2009; Kumaran & McClelland, 2012) and imagination (Hassabis et al., 2007).

**Transformers, context, and episodic memory in natural intelligence:** Several recent works note related connections between key features of transformers and natural intelligence. Most saliently, Russin et al. (2025) highlight how capacity and flexibility tradeoffs between parametric learning and in-context learning relate to similar tradeoffs in human intelligence—specifically, with in-context learning having greater flexibility and potential for compositional generalization, but less capacity, relative to in-weights learning. More generally, various works have noted connections between transformer attention mechanisms and episodic memory in the brain (Whittington et al., 2022; Gershman et al., 2025). Our work complements these, by showing how these connections could point to important contributions of episodic memory to generalization, in both transformers and natural intelligence.

## 2.2 Artificial Intelligence

**Prospective learning:** The topic of latent learning relates to the argument of De Silva et al. (2023) that machine learning assumes that the future will be identical in distribution to the past, but natural intelligence learns more prospectively, in a way that is oriented towards generalizing in the future. The challenges of latent learning can be interpreted as a type of prospective learning—learning information not because it is useful under the current task distribution, but because it might be useful if the task distribution changes in the future. However, the type of prospective learning focused on by De Silva et al. involves explicitly modeling how tasks might change in the future; retrieval avoids this challenge by leaving the question of which tasks may be relevant to the future (see below).

**Meta-learning and in-context learning in transformers:** Another way a system can adapt to a novel future task is by learning that task once it appears; many works have explored how this capability can be meta-learned (e.g., Santoro et al., 2016; Wang et al., 2017), often using context in transformers as a form of memory (Parisotto et al., 2020; Lampinen et al., 2021). Many works have similarly interpreted the emergent in-context learning ability in language models (Brown et al., 2020) as a type of implicit meta-learning over the training distribution (Xie et al., 2022; Chan et al., 2022a; Lampinen et al., 2024). Thus, there has been substantial interest in meta-learning (across contexts) to learn how to use information *within* a single context. However, meta-learning alone does not fully address how knowledge from past contexts could be directly applied at present.

**Using memory of prior contexts in agents & language models:** Several prior works in RL focused on the benefit of retrieving memory of past contexts to help adapt to present ones (e.g., Ritter et al., 2018; Pritzel et al., 2017; Goyal et al., 2022; Humphreys et al., 2022). More recently, there has been substantial interest in retrieving related training documents to improve test time performance in LMs. Seminal work on Retrieval Augmented Generation (RAG; Lewis et al., 2020) showed that retrieving Wikipedia articles, and adding them to the context[1], could substantially improve language model responses. More recent works have used retrieved documents to fine-tune the models (Hardt & Sun, 2024; Hübotter et al., 2025), explored

---

[1]The full RAG strategy involves additional marginalization over multiple documents, etc.; however, each document is effectively just prepended to the context.

incorporating retrieval into the context throughout training (e.g., Khattab & Zaharia, 2020; Borgeaud et al., 2022), or allowed active retrieval via use of tools like web search (e.g. Schick et al., 2023).

However, these works have generally *not* focused on which particular modes of generalization would be supported by retrieval vs. parametric learning; indeed, in many cases the retrieved examples might contain the test answer verbatim, and fine-tuning on them would work equally well. Our work highlights the unique role that retrieval can play in ameliorating certain failures of generalization from parametric learning alone—and provides a complementary perspective on understanding the benefits of common techniques like RAG.

**Patterns of generalization in language models:**   Several streams of work identifying failures and successes of generalization in LMs helped inspire our investigations. First, the Reversal Curse (Berglund et al., 2024), and failures of multi-hop inference (Balesni et al., 2024), are intriguing failures of LMs to fully exploit the information available in their fine-tuning data. Other studies similarly find inadequate cross-lingual transfer of knowledge in language models (Qi et al., 2023; Aggarwal et al., 2025; Goldman et al., 2025). We interpret these findings as instances of the broader phenomenon we highlight: language models learn what is explicit in the training data, in ways that may not generalize to all its latent implications for other questions (see Fig. 1b).

However, other works have found instances of surprisingly-*successful* generalization in language models. For example, several works demonstrate cases where models seem to generalize to a test example based on facts stated in other training documents (Berglund et al., 2023; Meinke & Evans, 2023). Similarly, Cook et al. (2025) show that models can generalize from code for functions in training to execute those functions at test time. These examples show cases where language models *do* exhibit surprising generalization "out-of-context" in a way reminiscent of latent learning, though the absolute generalization performance observed is usually low. We return to this issue in the discussion.

**Generalization from fine-tuning and in-context learning:**   Several closely-related works show differences in generalization between in-context learning and parametric learning in pretrained language models (Lampinen et al., 2025; Park et al., 2025; cf. Chan et al., 2022b). Like our results, these works show that context-based learning can offer unique contributions beyond parametric learning. However, they focus on pretrained language models and narrower evaluations, and do not clearly identify the division between explicit and latent knowledge that we highlight here. Many of these works propose data-augmentation-based solution, as do other related approaches (Akyürek et al., 2024; Yang et al., 2025; Ruan et al., 2025). Augmentation can enable expending additional train-time inferences to improve test-time generalization by filling in the latent information; however, this requires the model (or model trainer) to infer *a priori* which latent tasks should be targeted for augmentation. Our results show an alternative perspective by which retrieval from episodic memory can allow performance *without needing to decide at training time which tasks may be relevant later*—overcoming similar challenges while remaining more flexible. We return to these issues in the discussion.

## 2.3   Similarity-based and procedural generalization from parametric learning

As noted above, we want to contrast the type of latent learning challenge that we describe here with the well-documented strengths of generalization from parametric learning. There is substantial evidence that models can and will generalize—even out of distribution—under controlled tests (e.g., Abreu et al., 2025; Misra & Mahowald, 2024). We highlight two such modes of generalization here.

**Similarity-based generalization:**   First, there are many studies showing a kind of *similarity-based* generalization from parametric learning. When entities share common features, parametric learning will build representations that reflect that similarity (Hinton, 1986)—and even capture structures such as analogies on the basis of similar relations (Mikolov et al., 2013). These structured representations can enable some of the generalizations characteristic of human semantic cognition (Rogers & McClelland, 2004), and likely underlie many of the successes of modern machine learning. We will show below that, in the presence of the right similarity cues, this type of learning can allow successful generalization in some cases that would otherwise require latent learning.

**Procedural generalization or learning functions:** The second mode of generalization that parametric learning can enable is learning generalizable *procedures* or *functions* over structured inputs, that can be correctly applied in a new situation. Again, there is a long history of connectionist work studying this capability. For example, connectionist models can learn a mapping from the present tense form of a word to its past tense, in a way that generalizes to items never seen in training (Rumelhart & McClelland, 1986) and handles both regular and exceptional items gracefully (McClelland, 2015). More recently, there has also been ample evidence of stronger structural generalization from deep learning models—even in areas that have been considered challenges, such as learning generalizable procedures for variable binding (Hill et al., 2020), compositional generalization (Lake & Baroni, 2023) or causal reasoning (Lampinen et al., 2023). The flexible in-context reasoning procedures that our models demonstrate here are an instance of procedural generalization.

## 3 Towards a formal framework for latent learning

We next sketch a framework in which we can begin to formalize some of the ideas outlined above about the distinction between latent learning and other types of generalization. Note that this description is intended to illustrate the high-level abstractions, not to fully characterize all the forms in which a latent learning challenge may occur in practice; we discuss limitations at the end of this section.

Let $X$ be an input space, and let $X^*$ denote the set of finite sequences of elements from that space. Let $T^*$ similarly be a space of task-cue inputs. Suppose that a model is provided an input sequence $x = [x_0, ..., x_k] \in X^*$ and a task cue ($t \in T$) and is trained to *both* causally reconstruct its inputs and task cue *and* reproduce a task-cue-modulated mapping to an output space $f : X^* \times T^* \mapsto Y$. For example, the model might be a transformer trained via autoregressive prediction on sequences of the form $[x, t, f(x, t)]$. In a language processing setting, the input sequence $x = [x_0, ..., x_k]$ might be the concatenation of the individual sentences ($x_i$) that make up a document, the task cues $t$ might be questions about that document, and the mapping $f(x, t)$ would produce the correct answers to those questions. In an agent setting, like the latent learning experiments in rats, the inputs $x$ might consist of the identity of a maze (e.g., its name) together with the sequence of the agent's observations and prior actions, the task cue $t$ might consist of a navigation target (e.g., a particular object in the maze), and the mapping $f(x, t)$ might consist of the optimal next action to take towards that goal.

Note that $f$ may be non-injective; in particular, we assume that given a task cue $t$ there may be other related sequences of inputs $x' = [x'_0, ..., x'_{k'}]$ that produce the same output $f(x, t) = f(x', t)$. For example, in the maze setting the agent may end up at the same location by different paths (or even start a fresh episode at that location); whatever its history, the optimal action for navigating to a given goal will be the same. Or, in the language modeling setting the answers to some questions (e.g., "Who taught Aristotle?") are the same even without the relevant information in context, or with the information in a different format. This redundancy opens the possibility of generalizing knowledge across different input sequences. However, auxiliary information in context, such as the previous objects seen in the maze, or the relevant information encountered in a document, may help the agent to solve a task that it has not yet (fully) learned.

Suppose that in training the model encounters a particular input sequence, task, output tuple $[x, t, f(x, t)]$. Suppose that there is another possible task $t'$ that could be applied to the same (or similar) inputs, with a corresponding output $f(x, t')$—and that the model has learned the task $t'$ from other training examples. Our claim is that even if the model were to predict $t'$ as a likely task cue given the input sequence $x$, and even if the model could correctly produce $f(x, t')$ if cued with $t'$ in context, parametric learning on the sequence with the original task cue $t$ will not in general encode the alternate possible sequence with $t'$—or generalize to the other sequences $x'$ in which that learning could be applied.

We would like to distinguish this from many types of generalization that a sufficiently expressive and implicitly/explicitly regularized model *will* acquire from parametric learning over sufficient data. First, the model will learn transferable procedural knowledge about the tasks: if there is a systematic input-output relationship for task $t$, models will tend to learn the mappings in a way that generalizes $t$ to novel inputs that provide sufficient information in context (subject to the usual constraints on generalization in machine learning, e.g., under distribution shift). Second, the model will learn the associations and similarity relation-

ships among the inputs, task cues, and outputs. In some cases, this may allow improved performance even on input, task pairings $x', t'$ not seen in training; as a trivial example if for a given input $x$ the output is always constant ($\forall t \in T\ f(x, t) = c$), the model will not need to see all possible tasks to learn this. Similarly if multiple task cues are redundant, if certain tasks only have a subset of possible outputs, etc., the model may readily learn these regularities.

Nevertheless, we will show that there are cases where the latent task $t'$ is important, but is sufficiently different from the information contained in the inputs or the outputs for the learned task $t$ that the model will fail to learn it parametrically; e.g., the answers to reversal questions about a document, or navigating to alternative goals in a maze. In these cases, we will argue that episodic memory—by reinstating the full original input context $x$—can enable more flexible generalization from training experiences than parametric learning alone.

To make that more concrete, consider the following conditions and the performance we hypothesize for each—with examples from the case of a reversal (cf. Fig. 1a). Let $x = $ "Plato taught Aristotle" and $t = $ "Who did Plato teach?" Let $x'$ be an empty context, and $t' = $ "Who taught Aristotle?". Then the performance we predict is:

- **Forward with context $[x, t, f(x, t)]$**: E.g., "Plato taught Aristotle. Who did Plato teach?" "Aristotle." This condition corresponds to the forward direction seen in training, and can be solved either by using the in-context information or by recalling the forward fact. The model will perform well at producing the final answer.

- **Forward without context $[x', t, f(x, t)]$**: E.g., "Who did Plato teach?" "Aristotle." Although the information is not given in context, models will generally be able to give the final answer correctly after having been trained on examples like the above.

- **Reversal with context $[x, t', f(x, t')]$**: E.g., "Plato taught Aristotle. Who taught Aristotle?" "Plato." Although this question involves a reversal of the trained relation, because the forward direction is present in context the model will solve it correctly. Together with the previous item, this shows that the answer to the reversal question is *latent* in the information learned by the model—it has learned all the pieces it needs to solve the problem.

- **Reversal without context $[x', t', f(x, t')]$**: E.g., "Who taught Aristotle?" "Plato." The crucial test of latent learning. The necessary information has been learned, and the question could be answered with that information in context (the two conditions above). However, we claim that a causal[2] language model will generally *not* answer this condition correctly from parametric learning alone.

Analogously, an agent conditioned on a sequence of observations $[x_0, ..., x_k]$ produced by pursuing a particular goal, with a task cue $t$ specifying that goal (e.g., "Go to the blue pyramid"), will not learn about the path to take to reach other objects for goals $t'$ ("Go to the red sphere") even if it estimates those to be likely goals in the maze and could readily solve those tasks given the present observations. These possible counterfactual goals $t'$ and outputs are *latent* in the inputs, but because they are not cued and executed, there is no direct pressure for the model's parameters to learn to encode them in a way that will be useful if they *are* cued in the future.

It is these crucial types of latent generalization where we argue that episodic memory can have the largest effect by bridging the latent learning gap—effectively converting the difficult final condition into the easier ones that are solvable by in-context learning.

As noted above, this formal is more limited than the full scope of settings in which latent learning challenges occur; for example, in language modeling there is usually no complete disentanglement between the observa-

---

[2]Of course, the reversal failure is specific to causal language models; models that do bidirectional prediction over the same data would not exhibit it (Wu et al., 2024). This finding is compatible with the perspective expressed here, because changing the prediction task to be bidirectional makes the reversal information more explicit in the training task. This observation illustrates the broader point that what is "explicit" or "latent" in a dataset is not purely a function of the data, but also the architecture and objective used to train it. Nevertheless, we hypothesize that bidirectional models would still exhibit the other, more complex latent learning challenges we describe, e.g. in the next paragraph.

tions $x$, the task indicators $t$ and the outputs $f$ to be learned. This entanglement also occurs in some of our examples, e.g., it is difficult to fully separate observations from task indicators in the gridworld environment. Likewise, there are other forms of redundancy in $f$ (aside from non-injectivity), that could enable latent generalization. We leave fully characterizing the extent of latent learning to future work. However, we hope that this formal statement makes it more clear where systems might suffer from a challenge.

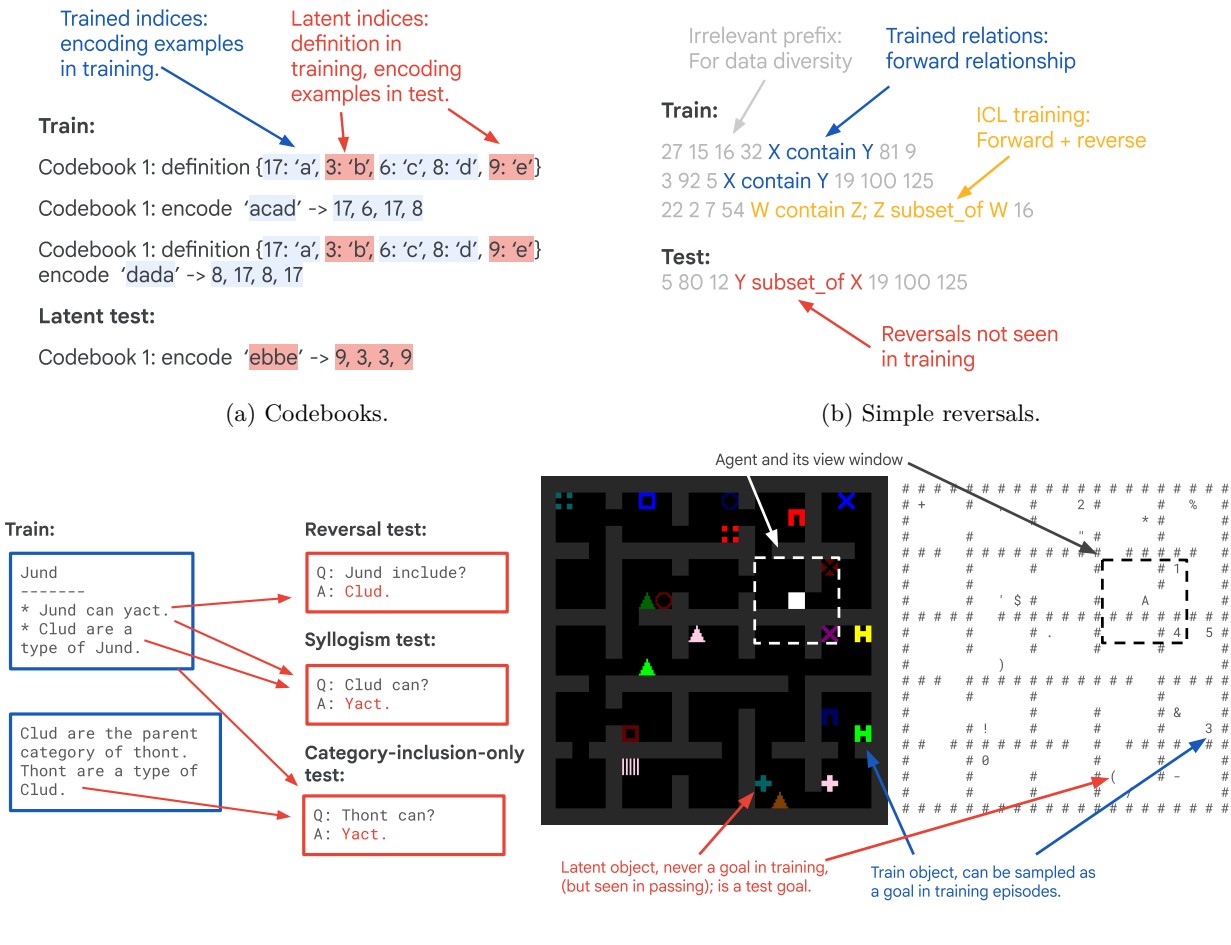

(a) Codebooks.

(b) Simple reversals.

(c) Semantic structure.

(d) Gridworld (pixel & ASCII versions).

Figure 2: The benchmarks we use and the key types of latent generalization that they test. (a) The codebooks benchmark tests the ability to use latent indices (highlighted in red) for which only the definitions have been seen in training to complete test encoding sequences. (b) The simple reversals benchmark tests the ability of models to reverse relations seen in training, and which models have learned to reverse in-context. (c) The semantic structure benchmark uses training embedded in more naturalistic text to test latent generalization types ranging from reversals to syllogisms, or more challenging category-inclusion-only holdouts. (d) The latent gridworld—with both its pixel-based RL and ASCII-based BC instantiations—tests the ability to navigate to objects that have never been a navigation goal in training for a particular maze, but have been frequently seen. (The same maze is shown in both pixels and ASCII; the agent's view window is shown with a dashed line for clarity.)

## 4 Benchmarks

We evaluate latent learning and the benefits of retrieval across a broad variety of settings (Fig. 2), ranging from recalling and using codes, to reversals and factual reasoning, and navigating to goals in a maze—and with both supervised and reinforcement learning algorithms. We begin with two benchmarks (codebooks &

simple Reversals) that exemplify the distinction between latent learning and other kinds of generalization in simple settings, and then expand to two more complex benchmarks that involve richer structures and tasks (semantic structure & latent gridworld). Across this broad range of settings, we demonstrate how parametric learning accurately acquires and generalizes certain aspects of the tasks, yet fails to learn *other* information or goals that are latent within the setting—leading to key failures on tests of latent learning. In this section, we briefly describe these benchmarks; see Appx B.1 for full details.

### 4.1 Codebooks

The codebooks benchmark is a simple demonstration of the failures of latent learning, despite successful generalization of various other types. The dataset is generated from a large number of codebooks. Each codebook consists of a mapping from a fixed set of 40 input tokens (shared across all codebooks) to a larger set of 128 possible output tokens.

Each document begins with the unique codebook identifier and then has either the definition of the codebook, or an encoding task using the codebook, or both (see the train set examples in Fig. 2a for an illustration of each type of sequence). The definition consists of a list of mapping pairs in key-value format, like a Python dictionary. The encoding sequences consist of a task cue indicating encoding and a "plaintext" string of input tokens to be encoded; the output is the string encoded via the current codebook. Each portion of a sequence is separated by a unique delimiter cue indicating the type of content to follow (e.g., before a codebook definition there is a "<definition>" token).

We generate a large set of training sequences of each type from these documents. Crucially, for a subset of the codebooks (the "latent codebooks"), we hold out some of the codebook input/output pairs from being used in the training encoding sequences. These pairs still appear in the definition sequences (including the definition portion of definition and encoding sequences), but the training set does not include any examples of their use for actual encoding (red indices in Fig. 2a).

We evaluate the model on a variety of validation sets intended to measure its recall of the training information (i.e., can it recall a codebook definition when cued with its identifier), its ability to encode novel sequences from the trained codebooks (for non-latent indices), and its ability to do things like ICL (learn a new codebook in context, and then apply it to encode a novel sequence). The crucial test, however, is of *latent encoding*: an encoding sequence (evaluated without the definition in context) consisting solely of held-out indices from one of the latent codebooks.

Our claim will be that although the model will pass most the above tests, it will fail on the crucial latent encoding test. That is, the model cannot use the information latent in the training set to encode a novel sequence using indices that were not used in encoding examples for that codebook during training—despite already "knowing" the necessary information, in the sense that it can recall the codebook definition correctly, and can successfully execute encodings on the held-out tokens with the definition in context. This will show that even when the model learns the codebook definition, and how to encode from a definition, it is failing to learn the latent information (i.e., the unused input-output pairs) in a way that generalizes without the definition in context.

### 4.2 Simple reversals

This dataset is loosely inspired by the reversal curse dataset of (Berglund et al., 2024), and the simple reversals dataset of Lampinen et al. (2025), but is designed to have sufficient diversity for (minimal) training from scratch. To do so, we generated a set of facts about 1,000 entities and 20 relations (plus a reverse relation for each, for a total of 40). e.g., if X and Y are entities, a relation might be "X contains Y" and its reverse might be "Y subset_of X." In this dataset, the relation token, and the in-context reversal cue, can each be seen as task cues that indicate what type of output should be produced. We train the model on a set of 20,000 relations in the forward direction, and 19,800 of their reversals, holding out 200 reverse relations for a test set. To create training documents, we augment each relation sentence with up to 5 random prefix and suffix tokens. We also include 12.5% of ICL sequences that have both forward and backward relations

in the context. We validate generalization to the trained forward relations with novel prefixes, and test generalization to the held-out reversals.

### 4.3 Semantic structure dataset

This dataset is adapted from the semantic structure benchmark of Lampinen et al. (2025); however, as we are training models from scratch rather than finetuning, we make a few modifications. We generate data using the same generating process from a larger structure of 1,100 entities, sample a larger number of documents (11,000), and tokenize the dataset such that each word (or punctuation mark) is represented by a single token. We evaluate performance on the same evaluation types as the original work: simple rephrasing (without changing relation order), reversals, syllogisms (two-step logical inferences), and category-inclusion-only holdouts (holding out all facts about a category except its parent, testing on all inferrable facts). These tests identify different ways of flexibly using information latent in the training data. We evaluate using multiple choice questions, but we increase the number of choices to 64 (up from 4 in the original work) to make the questions more difficult. As noted in the description above, the boundaries of a "task cue" are blurrier for this dataset—especially in training—but in general for evaluation the questions can be seen as a task cue which should induce an appropriate answer as output.

We also create two versions of the dataset. In the first, there are strong similarity-based cues that can provide shortcuts to inferring an answer. For example, if there many birds—all or most of which share properties like flying, nesting, having wings, having hollow bones, etc.—even if the particular statement "eagles have wings" is held out, the model may be able to infer it via similarity-based generalization from other birds (e.g., Rogers & McClelland, 2004). We create another version of the dataset in which these similarity-based cues are reduced, by making the semantic structure tree wider and shallower (more heavily branching at the top, so that fewer leaf nodes have common properties) and choosing the test and the distractor choices from the same branch of the tree (so that all the possible choices have many similar features). We use these two versions to show how similarity cues affect parametric and episodic generalization.

### 4.4 Latent gridworld navigation task

This task is inspired by the original latent learning experiments in rodents (Tolman, 1948). It involves navigating through gridworld mazes (Fig. 2d), and learning about explicit and latent goals within them. In each episode, the agent is placed in one of a set of $N$ fixed mazes, each of which consists of a grid of rooms, with each room being surrounded by walls with occasional doors. The agent is given a limited view window of $5 \times 5$ squares centered on its current position (unaffected by surrounding objects). The task for the agent is to navigate to one of 20 objects scattered throughout the maze given an instruction (i.e., a task cue). The maze environment is based on the Zipfs Gridworld environment (Chan et al., 2022c).

Within each maze, 15 of the objects are used as navigation targets in training, while 5 are held out as latent objects. A different set of objects is held out for each maze. The validation tasks consist of navigating to the trained objects from locations in the maze that were not a starting point in training. The latent test evaluations consist of navigating to the 5 held out objects; although the agent has not been explicitly trained to navigate to these goal objects, like the rats in the latent learning experiments, it will have passed by them many times during training. We experiment with two different settings where agents are trained either with reinforcement learning (RL) from pixels or with behavioral cloning (BC) from an ASCII representation.

**RL** We sample a larger number of mazes according to a skewed, Zipfian distribution and add an in-context leraning structure to the environment within each task by having the agent navigate to 5 subgoals within the maze. As in prior emergent ICL work (Chan et al., 2022a), the presence of bursty and long-tailed learning experiences incentivize ICL while remaining ecologically plausible.

**BC** We sample uniformly from a smaller number of mazes. To move closer to the language-modeling setting, we train agents with behavioral cloning on optimal trajectories by predicting the next token in a sequence that interleaves environment observations and optimal actions and evaluating it via online (interactive) performance. Observations consist of flattened ASCII grid representations, augmented with maze index, and navigation target cues.

## 5   Methods

We train our models using relatively standard architectures, methods, and hyperparameters derived from prior work; we present full details in Appx. B.3. For the language-like tasks (i.e., all except RL Gridworld), we train a decoder-only transformer on a standard causal language-modeling objective using ADAM (Kingma & Ba, 2014). For the RL tasks, we train the agent using IMPALA (Espeholt et al., 2018)—including auxiliary reconstruction losses that force the agent to reconstruct its visual and textual inputs (cf. Chan et al., 2022a). Thus, in each case the model is being trained to reconstruct the full information in its training experiences, rather than simply ignoring some portions.

**Oracle retrieval**     For the oracle retrieval condition, we gave the model (agent) access to at least one relevant document (episode) that contained information relevant to solving the task, by prepending it to the context, along with some irrelevant distractor episodes sampled uniformly, except for the BC Gridworld benchmark where only relevant trajectories were provided. The total number of retrieved episodes (including distractors) varied from 3 to 7 across the tasks. In the supervised environments the retrieved episodes are re-encoded; however, in the RL environment they are retrieved as cached memory states.

When training with oracle retrieval, we do *not* train the model (autoregressively or otherwise) to predict the retrieved information for the current batch; it is provided only as context (though gradients are propagated into this context). Thus, after a fixed number of training steps, the *number* of tokens on which the model will have received a loss signal is identical between the retrieval model and the baseline. (See also the ablations below.)

## 6   Results

The results are organized as follows. First, we show how baseline transformers model exhibit striking failures to exhibit latent learning on some of our key datasets—despite achieving high generalization performance on various other test sets. These results motivate the introduction of oracle retrieval; which we show resolves these issues. We then expand our focus to study these results in the context of the Semantic Structure benchmark—showing the additional effects of associative cues, and weaker benefits without strong ICL examples in the training data. Finally, we turn to the Gridworld environment—in both its pixel-based RL and ASCII-based BC instantiations—and show how in both versions, retrieval substantially improves over parametric learning alone.

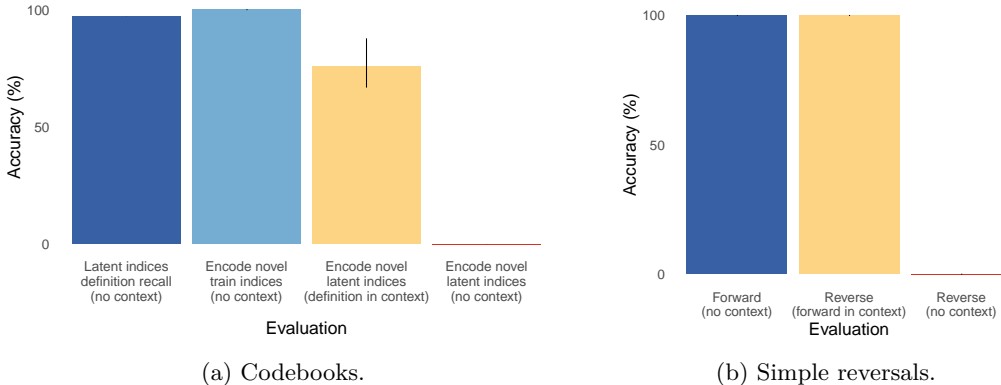

(a) Codebooks.                    (b) Simple reversals.

Figure 3: The inability to latent learn demonstrates the potential benefit of episodic memory—models often contain the information they need to solve a task, and can solve the task if that information is in context, but cannot put the pieces together to achieve latent learning. In each plot, the right-most bar (with close to zero performance) is the latent test; the other bars are the pieces needed to solve it: the ability to recall the relevant information (blue bars) and the ability to use the relevant information to solve the task in context (yellow bars). (Errorbars are 95%-CIs calculated across 4 runs; where errorbars are omitted, the rates were identical across runs.)

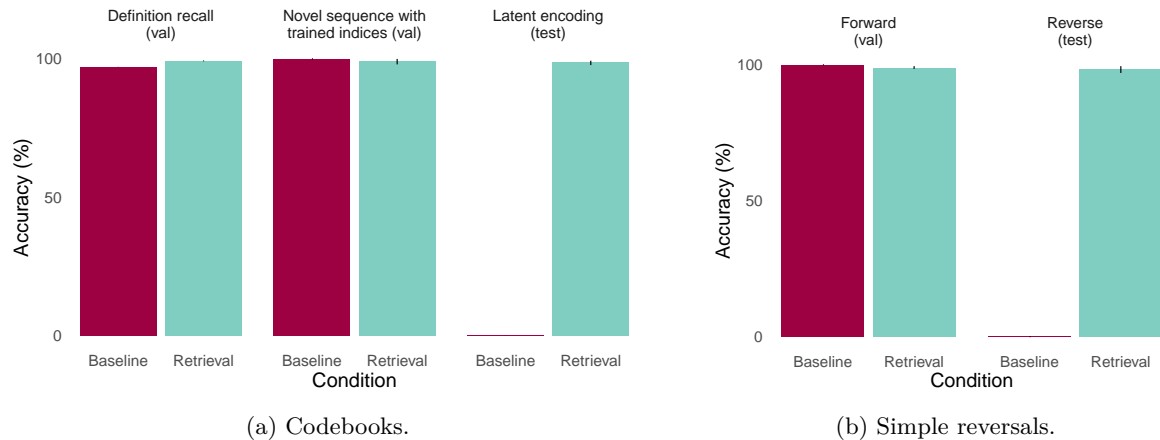

(a) Codebooks.

(b) Simple reversals.

Figure 4: The benefits of oracle retrieval on the (a) codebooks and (b) simple reversals benchmarks. Both baseline and retrieval models perform well on component tasks like recalling definitions, or encoding new sequences involving indices used in encoding during training (a, center). However, performance differs dramatically on the latent encoding test (right bars on both plots), where only the model with retrieval achieves above-chance performance. (Errorbars are 95%-CIs calculated across 4 runs; where errorbars are omitted, the rates were identical across runs.)

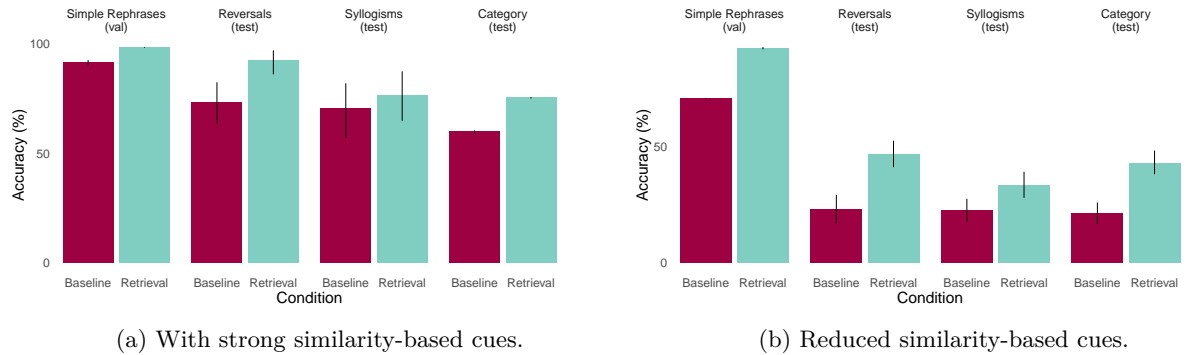

(a) With strong similarity-based cues.

(b) Reduced similarity-based cues.

Figure 5: Results on the semantic structure benchmark, comparing performance of a baseline model and one with oracle retrieval, in settings with and without strong associative cues. (a) When strong similarity-based cues are present in the data, both models achieve relatively high performance due to the possibility of associative learning. This demonstrates how similarity-based generalization can provide an alternative route to generalization in some cases. (b) When similarity-based cues are reduced, the advantage of the retrieval model is more notable. However, in all cases the benefits of retrieval are more muted than in other cases, likely because there are not sufficient examples for the model to acquire strong ICL capabilities. (Errorbars are 95%-CIs calculated across 4 runs; where errorbars are omitted, the rates were identical across runs.)

**Failures of latent learning in baseline models:** In Fig. 3, we first demonstrate the striking failure of the baseline transformer model to exhibit generalization to the latent test conditions on our simple reversals and codebooks benchmarks, despite readily passing the generalization conditions that would be needed to solve these tasks. These results motivate the idea that an alternative mechanism might be beneficial.

**Adding retrieval:** In Fig. 4, we show that oracle retrieval could solve the latent learning problems described above. On both the Simple Reversals and Codebooks benchmarks, a system equipped with retrieval not only solves the same validation tasks as the baseline model, but also generalizes very well to the key latent test conditions. This shows the striking difference in parametric and retrieval-based generalization on the latent learning tests.

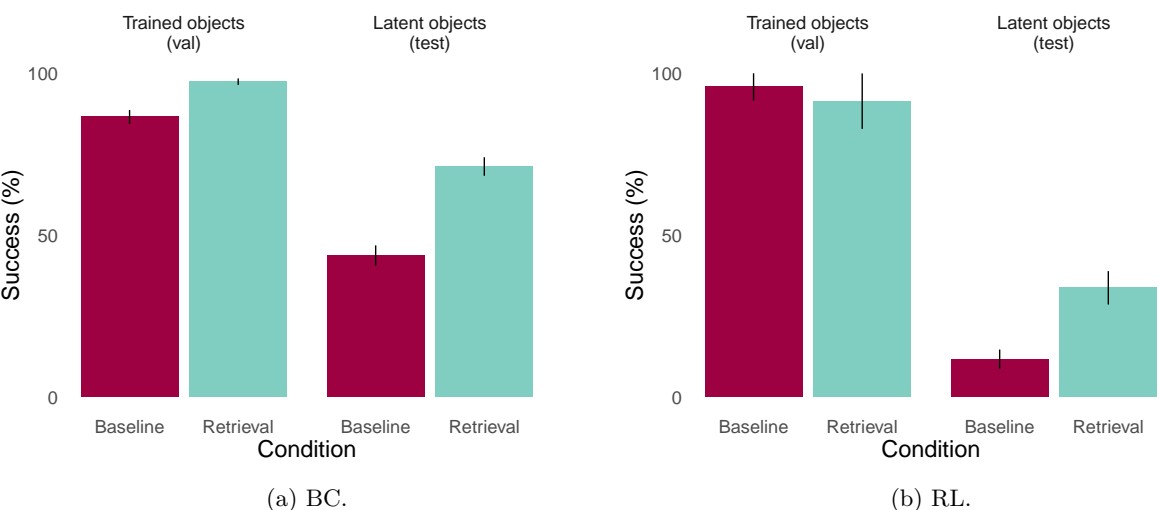

(a) BC.                (b) RL.

Figure 6: Results on the gridworld environment, comparing performance of a baseline model and one with oracle retrieval on objects latent in the most frequent mazes. Although performance on the latent objects remains much lower than validation performance, the agent with retrieval achieves significantly higher performance than the baseline agent on the latent object tests in both the (a) BC and (b) RL versions. (Note that some performance is achievable by purely exploring the maze, which is why even baseline performance is not as low as in the text-based experiments above. For BC results, errorbars are binomial approximation 95%-CIs; for RL they are bootstrap CIs across 3 runs.)

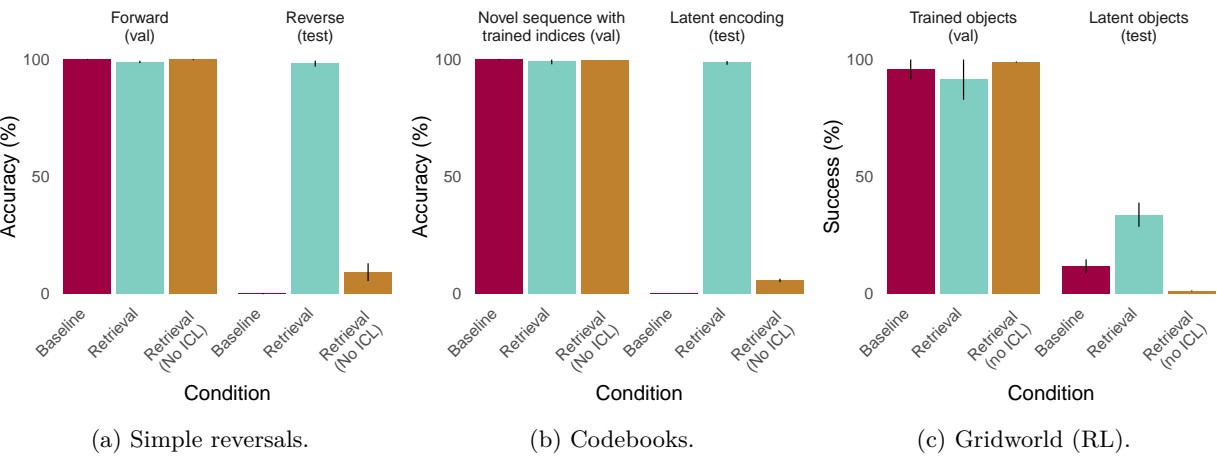

(a) Simple reversals.     (b) Codebooks.     (c) Gridworld (RL).

Figure 7: Having ICL-supporting sequences (or episodes) in training is necessary for achieving strong benefits from oracle retrieval. The brown bars compare models trained *with* oracle retrieval, but *without* other ICL-supporting sequences in the data, to the Baseline & Retrieval results reproduced from above. (a) Simple reversal results. Despite having oracle retrieval at training and test, the model trained without explicit ICL sequences struggles on the latent test set, only achieving around 12% performance. (b) Codebooks results. Again, Retrieval performs well on validation sets even without ICL examples in training, but struggles with the key latent tests, achieving around 6% performance. (c) Analogously, an RL agent trained with Retrieval, but using only single-task episodes that remove the potential for within-episode ICL, performs well on trained objects, but fails on the latent evaluation tasks. (Errorbars are 95%-CIs calculated across 3 runs; where errorbars are omitted, the rates were identical across runs.)

**Semantic structure benchmark:** We next explore two versions of the more complex semantic structure benchmark (Fig. 5). These experiments demonstrate several interesting effects. First, in the presence of

strong similairty-based cues, even the baseline model can generalize fairly well to holdouts like reversals (though note that the benefits from retrieval are still statistically significant at the $p < 0.05$ level for all conditions except syllogisms). These similarity cues are due to, for example, the fact that other entities similar to the one in question tend to have similar attributes. When these associative cues are removed from the dataset, however, performance of both models drops—and the benefits of retrieval become clearer. However, the benefits of retrieval still remain limited; one possible explanation is the fact that this dataset lacks within-document ICL examples from which the model could learn how to most effectively use memory (see below).

**Gridworld:** We finally show results on the more complex gridworld environment (Fig. 6). In both the RL and BC variants of this setting, we again see a significant improvement of oracle retrieval over the baseline model in generalization to latent goals in the environment. However, performance is still far from ceiling—perhaps reflecting the greater challenge of using memory (or memories) to identify and execute a long sequence of actions in pursuit of a novel goal, in contrast to the relatively more atomic facts that needed to be recalled in e.g., the Simple Reversals task above. Nevertheless, these results show how latent learning can be an issue for agents—even those trained with online RL—as well as other types of models.

**Within-example ICL helps learn to use retrieved memories:** We now highlight one result on how ICL interacts with retrieval. Specifically, we find that having training examples that promote *within-document* in-context learning (ICL) is important for achieving strong benefits of episodic retrieval (Fig. 7). To do so, we create a variation of the simple reversals dataset that omits the ICL-like sequences (i.e., the ones where a forward and backward sequence are included within the same document). Although the oracle retrieval model *still* experiences retrieved sequences that could be used to learn how to use memory, it in fact fails to achieve strong performance on the latent test split (though it does achieve non-zero performance, unlike the baseline model). These results show that learning to use context *within* a training example can support learning to use effective retrieval *across* examples. This finding may help to explain the weak benefits of retrieval on the Semantic Structure benchmark—where there are not clean ICL examples.

**Ablations:** We now briefly describe a few ablations (provided in full in the Appendix) that help to eliminate alternative explanations for our results, and clarify their limitations. First, in Appx. C.1 we show that the benefit of retrieval is not simply due to having more tokens in the batch, by showing that increasing the batch size—and even training on the extra tokens, unlike with retrieval—correspondingly does not result in improved generalization to the key latent tests. Second, in Appx. C.2 we show that the benefits of retrieval are not due to pure data augmentation, by showing that retrieving irrelevant examples does not substantially improve latent test performance. Finally, in Appx. C.5 we show similarly that the benefits of retrieval in the gridworld BC benchmark are not primarily driven by sequence length or batch size. Together, these ablations support our claim that the benefits we see from episodic memory retrieval are not primarily due to more prosaic features that improve generalization.

On the other hand, in Appx. C.3 we show that the *precision* of memory retrieval is important for the model to learn to use the memory effectively; while in our main experiments we always had some distractor memories, if we increase the number of distractor memories retrieved (while holding all other factors constant), the model eventually fails to learn how to use the memories effectively. This illustrates the challenge that must be solved to create an effective episodic retrieval system at scale.

## 7 Discussion

In this work, we have highlighted the challenge of latent learning—acquiring information that may be useful in supporting future tasks, in a way that can be flexibly used when those tasks arrive—as a potentially fundamental gap between natural and artificial intelligence. We have demonstrated this gap through both revisiting prior results, and proposing novel, controlled benchmarks. We then argued that episodic memory (as a form of nonparametric retrieval) could overcome these challenges; while parametric knowledge is relatively more fixed, retrieving information makes it available to the more flexible in-context reasoning processes of the system. To support this claim, we demonstrated that oracle retrieval could help overcome the latent learning gap—both in language modeling settings ranging from simple reversals to complex code tasks, and in reinforcement learning tasks inspired by the classic experiments on latent learning in animals.

We elaborated these experiments by identifying some of the features that support effective retrieval use (including within-experience ICL-supporting training), and ablation to eliminate other potential explanations. We close by returning to the themes introduced earlier in the paper, and discussing the broader picture of these results.

**Controlled settings for studying the surprising failures of language model generalization:**    As noted above, our experiments are inspired in part by a growing empirical literature that shows failures of large language models to make certain kinds of generalizations from their training data, including reversals (Berglund et al., 2024), syllogistic or multi-hop inferences (Lampinen et al., 2025; Balesni et al., 2024), or cross-lingual transfer (Qi et al., 2023; Aggarwal et al., 2025; Goldman et al., 2025)—as well as recent studies showing that models can generalize differently in context (Lampinen et al., 2025; Park et al., 2025). We study these findings in controlled settings that are isolated from confounding factors, and reproduce the qualitative phenomena, as well as shedding new light on the role retrieval could potentially play in resolving them.

**Understanding the empirical benefits of retrieval:**    One way of viewing our results is as offering an analysis of *how* retrieval may uniquely contribute to generalization, and thus *why* approaches like RAG (Lewis et al., 2020) may be beneficial. From this perspective, our experiments in controlled settings highlight that for certain classes of problems, retrieval may offer fundamentally different types of generalization that can complement parametric learning—and thus help to rationalize why retrieval is empirically useful.

**On generalization from parametric learning:**    As noted above, we are absolutely *not* arguing that models do not generalize out of distribution from parametric learning. Indeed, the flexibility of in-context learning to apply a procedure such as reversal or encoding to an instance that has never been seen in training is driven precisely by the procedural generalization of parametric learning. Our results on the semantic structure benchmark illustrate another mechanism—similarity-based generalization—through which certain kinds of generalization to systematic holdouts is possible, even in cases where latent learning might otherwise be necessary. Modes of generalization like these presumably explain many of the instances of successful generalization of modern machine learning. Thus, our claim here is simply that the type of latent learning we highlight in our controlled experiments is *one particular mode* of generalization that appears to be inadequately achieved by gradient-based parametric learning alone in present architectures, and that might be more effectively supported through episodic memory and retrieval.

However, as noted above, other works have found instances of surprising generalization from language models in similar settings to ours (e.g., Berglund et al., 2023; Cook et al., 2025; Meinke & Evans, 2023). For example, Berglund et al. (2023) show that models can generalize from training facts (in separate documents) like "The Pangolin chatbot answers in German" and "Latent AI makes Pangolin" to answer in German when prompted to answer as "Latent's AI." This finding may seem to show a type of latent learning that contradicts our claims above. However, in this case there may also be similarity-based or associative cues that are sufficient to give away the answer—even if it requires multiple steps of association (note that we observe some multi-hop similarity-based generalization on the syllogisms in the semantic structure benchmark). Moreover, the absolute degree of generalization observed in these studies is usually relatively low. Nevertheless, we believe that fully exploring the boundary between the types of generalization explored in this paper and other prior works is an important direction for future work. Factors like data diversity (along both relevant and orthogonal axes of variation) likely influence the exact structure of generalization.

**Complementary learning systems:**    In motivating this work, we highlighted that complementary learning systems theory (McClelland et al., 1995; Kumaran et al., 2016) emphasizes that episodic retrieval and parametric knowledge play complementary roles in natural intelligence. While past work in this area has often emphasized the benefits of episodic memory for learning from few exposures without interference, and assisting in integrating new information, our results highlight a different perspective; that episodic retrieval may unlock a fundamentally more flexible route to using information from past learning experiences, even if that information was not relevant to the task at learning time.

**Challenges of retrieval:**    Our experiments deliberately sidestep the difficult problem of *how* to do effective memory retrieval. However, as anyone who has struggled to recall something will know, even human memory cannot always recover relevant information at the time it is needed. While our experiments

show that in principle retrieval may unlock certain kinds of generalization, in practice the ability to use latent information in a training experience will be constrained by the ability to retrieve the correct experience from among the many that might be relevant. Thus, there will likely be a benefit to incorporating multiple, partially-redundant solutions to the latent learning problem.

**Multiplicity of solutions & online vs. offline replay:** We want to highlight that test-time (online) episodic retrieval is not the *only* solution to the types of problems that we explore. For example, several works have explored training-time (offline) augmentation to address challenges like reversals or multi-hop inferences (Akyürek et al., 2024; Lampinen et al., 2025; Yang et al., 2025). Other work on "preplaying" routes towards possible alternative goals while learning to navigate a new environment (Carvalho et al., 2025) similarly highlights ways to enrich learning such that future generalization will be improved. These solutions can be interpreted as applying effectively the same computational trick—using episodic retrieval to make latent information available for more flexible reuse—but doing so ahead of time to cache the solution for later. These training-time solutions therefore require more computational effort at learning time, but via this amortization avoid the retrieval problem at test time. On the other hand, these approaches also require estimating ahead of time which alternative goals may be relevant for future use; retrieval is more flexible. Thus, we expect that training-time augmentation methods likely complement test-time retrieval ones; ultimately natural intelligence is likely to rely on many strategies to improve its chances of generalization.

Indeed, there is substantial evidence that in natural intelligence, hippocampal replay occurs both online and offline, and plays multiple roles (Comrie et al., 2024; Eldar et al., 2020; Cowan et al., 2021). Online, replay helps to make predictions (Kay et al., 2020) and support flexible decision making (Eldar et al., 2020); offline, (p)replay contributes to consolidation and generalization (Ólafsdóttir et al., 2015; Kumaran & McClelland, 2012; Momennejad et al., 2018; Liu et al., 2019). We suspect that artificial intelligence will similarly benefit from employing multiple complementary approaches. Our goal here is simply to highlight that episodic memory retrieval can play an important role in various types of generalization, by bringing learning experiences back into context, where they can be used more flexibly to make inferences for the present or the future.

**Limitations:** There are a number of limitations to the present work—most notably the small scale of our experiments and our use of oracle retrieval. Our choice of these two factors has the benefit of improved experimental control—eliminating the risk of possible contamination of test examples from other training data, and isolate the effect of retrieval per se from the difficulty, but it is always possible that findings would be different at scale. This is particularly true with the unresolved problem of how to effectively do retrieval; while our experiments did not assume particularly high precision of retrieval (12.5% - 33% across experiments; other retrieved documents were distractors) further decreasing this precision eventually leads to failures, and we did assume perfect recall. Clearly, this standard would be difficult or impossible to achieve in a production-scale retrieval system. We leave addressing the question of how to practically resolve these challenges at scale to future work.

### Acknowledgements

We thank Sridhar Thiagarajan, Jörg Bornschein, Tyler Bonnen, and the anonymous reviewers for helpful comments and suggestions. We thank Murray Shanahan for support.

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

## A    Extended related work

In this section we provide some references to broader work relating to the themes of this paper, that did not fit naturally within the main text flow.

**Compression and generalization from parameters and context:**    A long-standing idea in machine learning is that compression is (at least an important component of) intelligence (e.g., Rathmanner & Hutter, 2011; Mahoney, 1999). This perspective has been applied to make sense of language modeling (Deletang et al., 2024) and in-context learning (Elmoznino et al., 2024) as effectively forms of compression—often as an explanation for their generalization. It is interesting to consider our results from this perspective. Our results highlight how the compression achieved through parametric learning may be more task- or format-biased than we would ideally want it to be, while models may have learned to use information more flexibly in context—precisely *because* that more flexible use results in greater compression of context-sequences over the training data. Episodic retrieval selects information to expose to these more flexible in-context processes, but exposes that information in a way that is less biased by the learning context.

## B    Detailed methods

### B.1    Benchmarks

In this Appendix we present more detailed descriptions of our benchmarks.

#### B.1.1    Codebooks benchmark

The dataset is generated from a large number (4,096) of distinct codebooks, each of which is identified by a unique index string. Each codebook consists of a mapping from a fixed set of 40 input tokens (shared across all codebooks) to a larger set of 128 possible output tokens (shared across codebooks, but because the set of output tokens is larger than the set of inputs, each codebook uses only a subset of the output tokens).

Each document in the dataset is a sequence generated using one of these codebooks. Every document begins with the unique identifier for the generating codebook. There are three types of possible documents:

**Definition:** the codebook identifier, followed by a list of the mapping pairs of the codebook in key-value format, like a Python dictionary.

**Encoding:** the codebook identifier, followed by a "plaintext" string of input tokens, followed by the string encoded via the current codebook.

**Definition and encoding:** as in the encoding sequence, except with the codebook definition included after the codebook identifier.

Each portion of a document is separated by a unique delimiter indicating the type of content to follow (e.g., before a codebook definition there is a "<definition>" token).

We generate a large set of training sequences of each type from these documents as follows. For each codebook, we add a definition sequence to the training set. We then add 64 encoding sequences and 64 definition and encoding sequences, for a total of 129 sequences per codebook. Crucially, however, for a subset of the codebooks (the "latent codebooks"), we hold out some of the codebook input/output pairs from being used in the training encoding sequences. That is, these pairs still appear in the definition sequences (including the definition portion of a definition and encoding sequences), but the training set does not include any examples of their use for actual encoding.

We evaluate the model on a variety of evaluation sets:

**Latent encoding test:** The key evaluation condition: an encoding sequence (without the definition) consisting solely of held-out indices from one of the latent codebooks.

**Latent definition recall validation:** A simple validation test that the model can successfully recall the definition of the codebook given its identifier (including the input/output pairs that are needed for the key tests above).

**Latent in-context encoding validation:** Validating that if the definition of the latent codebook is provided in context, the model can successfully execute the mapping even on sequences of held-out indices.

**Latent textbook trained indices validation:** Validating that the model can successfully execute an encoding mapping on a novel sequence using the *trained* indices from the latent codebook.

**In-context learning and encoding test:** Tests that the model can in-context learn an entirely novel codebook, and then use it to execute a mapping on a novel sequence.

### B.1.2   Simple reversals benchmark

We generated a set of facts about 1,000 entities and 20 relations (plus a reverse relation for each, so a total of 40 relations). Each entity is assigned to a single target for each of these relations. For example, one of these facts might be that "dax are fiercer than fep" and that would imply that "fep are meeker than dax." Thus, there are a total of 20 relations × 1,000 entities = 20,000 relations and 20,000 reverse of those in the dataset. Of these, the reverse pairs for 200 relations are held out to create a test set—only the forward direction is included in the training set. The question is whether being trained on these forward directions, along with the remaining 19,800 forward-backward pairs is sufficient to generalize to this test set. In order to add some variety to the sentences, each is repeated multiple times in the training data, with a random prefix and suffix token appended (sampled from a set of 100 tokens each, to avoid exact repeats). A small percentage (1/8th) of these are with the forward and reverse direction together in the same example (except for the sentences where the reversal is held out). The validation set consists of questions about the forward direction for the held-out reversals, but with different prefix/suffix tokens than in training; the test set consists of the reversals.

### B.1.3   Semantic structure benchmark

We generate data from a larger structure of 1,100 entities that is created by cloning the structure from Lampinen et al. (2025) 9 additional times, sample a correspondingly larger number of documents (11,000), and tokenize the dataset such that each word (or punctuation mark) is represented by a single token. We evaluate performance on the same types of test sets as the original work: simple rephrasing (without changing relation order), reversals of relations, syllogisms (two-step logical inferences over implicitly-quantified statements), and category-inclusion-only holdouts (holding out all facts about a category except its parent, testing on inferrable facts). Following the original work, we do systematic train-test splitting for each type of test that ensures the necessary information to an answer is present in the train set. We evaluate using multiple choice questions, but we increase the number of choices to 64 (up from 4 in the original work) to make the questions more difficult.

We also create two versions of the dataset; one in which there are strong associative cues that can provide shortcuts to inferring an answer (e.g., if all birds have wings, even if the statement "eagles have wings" is held out, the associations between "eagle" and "bird" may be sufficient to infer it), and another in which the associative cues are reduced, by choosing the tests and the distractor choices from the same branch of the tree (so that they share similar associative features aside from the key inferrable characteristics). We use these two versions to show how associative cues affect generalization.

## B.2   Latent gridworld navigation task

The maps are laid out in grids of 7 × 7 or 5 × 5 rooms for a larger or smaller map. Each room has 3 × 3 floor squares and is surrounded by wall squares which separate the rooms from each other and the edge of the map. Doors are randomly placed by replacing individual walls between the rooms with floor squares so as to ensure full connectivity, using the same algorithm as in the original environment Chan et al. (2022c).

The agent is given a limited view window of $5 \times 5$ squares centered on its current position with visibility not being affected by the type of surrounding squares, e.g., floor vs. wall. It can step in any of the 8 directions around it (i.e., it can take diagonal steps as well as axis-aligned ones). Below, we describe the differences between the RL and BC versions of this environment.

**RL from pixels**  We create 3,072 mazes with $7 \times 7$ rooms; mazes are sampled according to a skewed, Zipfian distribution (power law exponent $\alpha = 1$). To encourage the agent to learn how to use its memory, we add an in-context learning structure to the environment in training. Specifically, within each episode the agent is given a series of 5 navigation goals within the same maze (the goals differ, and the maze resets when the current goal is completed or a per-goal time limit is reached). Because of the long-tailed distribution of mazes, the agent will frequently find itself in unfamiliar ones; thus, it will be incentivized to learn how to in-context learn due to this bursty structure where it gets multiple tasks in the same maze.

**BC from ASCII characters**  We create 100 mazes with $5 \times 5$ rooms that are sampled uniformly. Agents are trained with behavioral cloning on optimal trajectories by predicting the next token in a sequence that interleaves environment observations and actions. Agents are evaluated in terms of task success by running them online to interact with the environment. Observations are represented by a flattened ASCII representation of the grid, where the agent, objects, and walls of the maze are distinct characters. Training sequences begin with tokens representing the navigation target and the index of the current maze, followed by interleaved observation and action tokens. For trajectories that are too long to fit into the fixed-size context of the transformer, we keep only as many of the initial observations and actions starting from the beginning of the trajectory as fit into the context. To ensure that retrieved sequences contain relevant information for the task, we filter retrieved sequences by map index and whether they contain the navigation object either in the target specification or the observations. Further, we found it necessary to also filter retrieved sequences to have the same or a nearby start location as the current agent location.

### B.3   Training

**Supervised**  For benchmarking on the supervised language-like tasks, we train a decoder-only transformer architecture (Vaswani et al., 2017)—generally with 12 layers, an embedding size of 1024, and 2048 hidden units in the MLPs on each task using the ADAM optimizer (Kingma & Ba, 2014).

The hyperparameters that differed across the supervised settings were respectively:

|  | Codebooks | Simple Reversals | Semantic Structure | Gridworld BC |
|---|---|---|---|---|
| Learning Rate | 1e-3 | 3e-5 | 2e-3 | 1e-3 |
| Seq. Len. | 256 | 32 | 128 | 1024 |
| Seq. Len. with retrieval | 2048 | 128 | 1024 | 4096 |
| Batch size | 1024 | 1024 | 1024 | 64 |

Table 1: Hyperparameter settings for the supervised experiments.

**RL**  We train our RL agenst using IMPALA (Espeholt et al., 2018). We generally follow the hyperparameters described by Chan et al. (2022a) in their experiments on a similar gridworld navigation environment (without the latent learning component). Importantly, this includes training all agents (including the no-retrieval baseline) with an unsupervised reconstruction loss that forces them to be able to reconstruct their visual and textual observations. We increased the batch size to 48, and changed the sequence length to 64, due to using different accelerators than the original work. Due to the larger size of our mazes, we also found it was necessary to increase the exploration-inducing entropy bonus to $3 \cdot 10^{-5}$ for faster learning.

**Oracle retrieval**  For the supervised domains, the episodes were re-encoded using the current model parameters; however, for the Gridworld RL case (due to technical obstacles) these retrieved episodes were presented in the form of the cached memory states created when the agent originally experienced the episode. Thus, the ability to use the retrieved episodes in the RL setting will likely be somewhat impaired by the

parameter drift of the model between the time of writing and retrieval. Nevertheless, we see significant benefits of retrieval.

## C    Supplemental experiments

### C.1    Ablation: the benefits of retrieval are not simply due to increased batch size

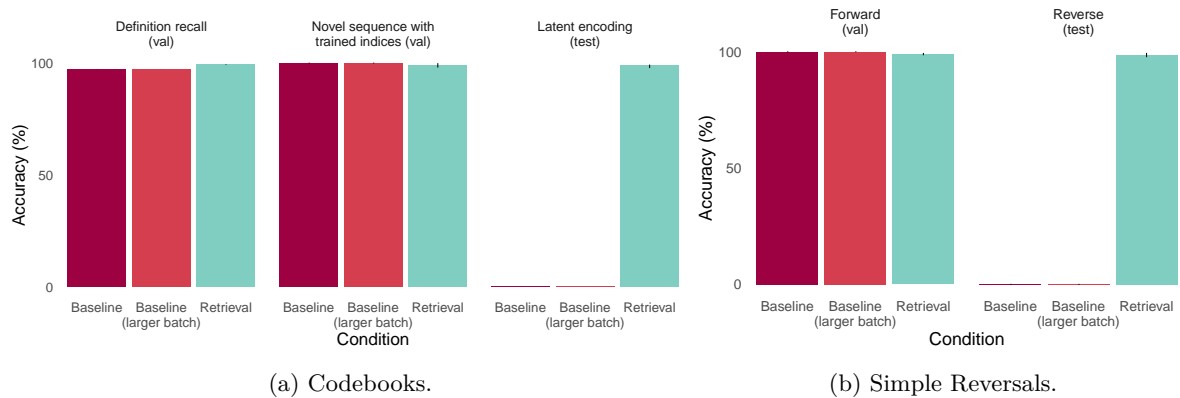

(a) Codebooks.                                                    (b) Simple Reversals.

Figure 8: The benefits of retrieval are not due to the (implicitly) augmented batch size; training a baseline model with an equivalently-much larger batch still does not result in generalization on the Codebooks or Simple Reversals tasks—despite the fact that it effectively increase the number of (non-unique) trained tokens dramatically compared to Retrieval.

Although we explicitly do not optimize the model's predictions on the tokens of the retrieved documents, it is conceivable that having those documents in context effectively increases the batch size of the model in some sense (e.g., by providing more optimization paths to the parameters in the *input* encoding of the retrieved documents). However, in Fig. 8 we show that the benefits of retrieval are not simply due to an implicitly increased batch size, by explicitly training an ablation model with a substantially larger batch size, but without retrieval. This model does not achieve any notable generalization to the latent test splits—despite having been trained on substantially more (non-unique) training tokens than the retrieval model.

### C.2 Ablation: the benefits of retrieval are not simply due to data augmentation

An alternative hypothesis would be that the benefits of retrieval are due to the increased context variability in which information is presented, which might tend to encourage encoding it more robustly. To test this alternative, we ran an ablation experiment where we trained a model with retrieval, but the retrieved documents were sampled to be *irrelevant* to the target one. This ablation resulted in no generalization to the key tests (Fig. 9), showing that the observed benefits of retrieval are not just due to context augmentation.

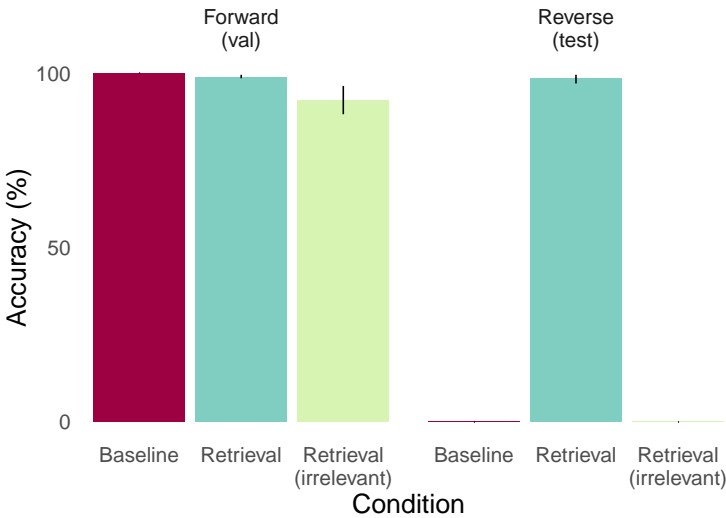

Figure 9: The benefits of retrieval are not simply due to data augmentation. If the retrieved information is from irrelevant documents (during training and test), retrieval no longer achieves improved performance on the key test conditions—and performance is even slightly impaired on validation examples in the same order. (Experiment on the Simple Reversals benchmark, cf. Fig. 4b.)

### C.3 Ablation: Retrieval precision must be sufficiently high for effective learning

In order to ascertain the importance of high-precision retrieval, we swept the number of irrelevant memories retrieved in the simple-reversals benchmark, while always keeping one retrieved memory relevant, and holding all other factors (dataset size, training epochs, learning rate schedule, etc.) constant. The results are shown in Table 2. At a fixed dataset size, performance quickly falls off as more irrelevant memories are retrieved, as the model starts to simply memorize the training examples rather than learning how to pull the correct memory out from amongst the many irrelevant retrieved memories. While with the larger datasets there might still be sufficient pressure to learn, this shows how memory retrieval precision can provide a fundamental bottleneck on learning to effectively use episodic memory.

| Number documents retrieved | Reversal test accuracy (%) |
|:---:|:---:|
| 0 | 0.0 |
| 3 | 99.5 |
| 5 | 67.2 |
| 7 | 0.8 |
| 9 | 0.0 |

Table 2: Simple reversals results with varying numbers of irrelevant retrieved memories (in addition to the relevant ones)—as the number of retrieved documents increases, the test performance falls off because the model fails to learn how to find the useful memories among the many distracting ones.

### C.4 Ablation: in the gridworld task, more maps are required to achieve generalization from RL than expert BC

There are a number of features that differ between the RL and BC settings of the gridworld. In the main text (ICL ablations) we ablated the multi-task episodes in the RL setting (which matches the BC setting), and showed that this strongly impairs latent test performance. Here, we ablate the other key feature that differs between the RL and BC gridworlds: the number of maps used in training. Specifically, in the main text RL experiments we trained agents over 3072 maps, while for the BC setting we used only 100. In Fig. 10, we show that using 100 maps in the RL setting strongly impairs test performance—reinforcing the point that there are interesting differences between the conditions required to achieve latent learning in the RL and BC versions of the environment. We speculate that these differences may be due to the distinct inductive biases of the supervised and RL losses, or the different representation formats, but we leave fully resolving the source of the differences to future work.

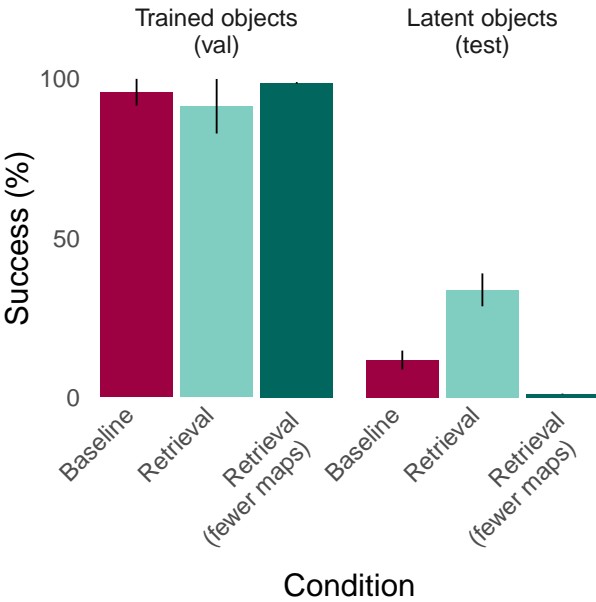

Figure 10: Ablating the number of training maps in the RL gridworld—with fewer maps, even a Retrieval model fails to achieve strong generalization to the latent tests.

### C.5 Ablating sequence length and batch size on the gridworld BC tasks

In Fig. 11, we show ablations where we vary the base sequence length for training, and then compare baseline and retrieval models with larger batch and sequence length ablations. As the base sequence length is increased from 256, to 512, to 1024, the base batch size is decreased from 512, to 256, to 64, respectively. Similar to the results above, we find that the benefits of retrieval are not primarily due to increased batch scope; except in the case of very short base sequence length, increasing batch size or sequence length offers little benefit while the effects of retrieval are much stronger.

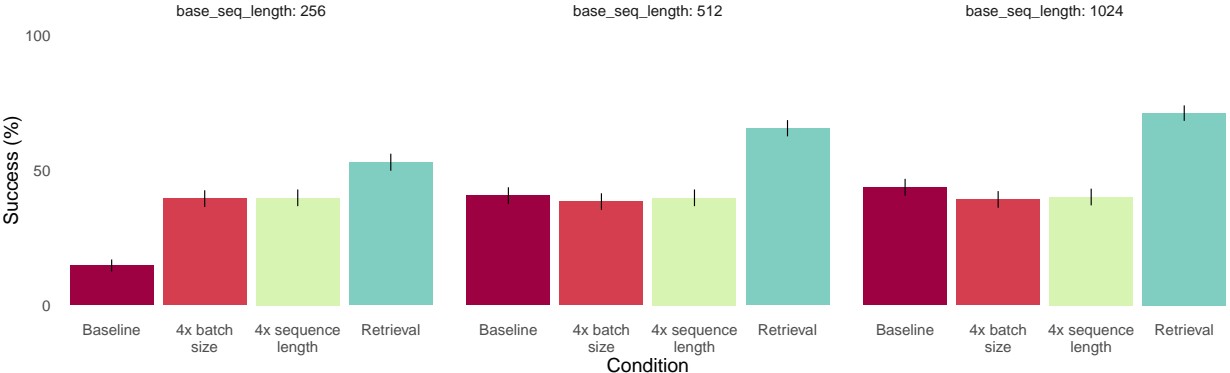

Figure 11: Ablating base sequence length and the benefits of larger batch size and sequence length for the benefits of retrieval in the BC gridworld benchmark: except in the shortest base sequence length condition (left), increasing batch size or sequence length offers little benefit over the baseline, while Retrieval offers large boosts in performance. (All results plotted on the latent test conditions.)

