# OpenReview forum: "Latent learning: episodic memory complements parametric learning by enabling flexible reuse of experiences"
_TMLR — Accepted by TMLR_

### Review · Reviewer_H96m · 2026-02-09

**Summary Of Contributions:**

The paper argues that many generalization failures in modern ML reflect a lack of latent learning: models often fail to flexibly reuse information that is present in training experiences but not demanded by the training task/format. Using several controlled benchmarks (codebooks, reversals, semantic structure, and latent gridworld), it shows that baseline parametric learners can solve the “latent” task when the relevant experience is in-context but fail without it, while oracle episodic retrieval (reinstating past episodes into context) largely closes this gap. It also finds that strong retrieval gains depend on training data that supports within-episode ICL, suggesting that effectively using retrieved memories is learned.

**Audience:**

No

**Audience Explanation:**

Although the topic is relevant, I do not think this submission would be of clear interest to much of TMLR’s audience because its main empirical takeaway is an oracle upper bound: if a relevant episode is handed to the model and prepended to context, latent-test performance improves. The paper does not address scalable storage/indexing, learned retrieval, or robustness to noisy retrieval, and most evaluations are highly synthetic/from-scratch, so the results provide limited actionable guidance for real systems beyond existing RAG intuitions.

**Claims And Evidence:**

No

**Claims Explanation:**

1. The work largely sidesteps the core difficulty of how to store/index and retrieve the right information at test time by relying on oracle retrieval, so it does not establish that the approach is feasible in realistic settings.
2. Most evaluations are on highly synthetic benchmarks, leaving unclear whether the phenomenon and gains transfer to naturalistic distributions.
3. The retrieval setup is overly “hand-holding,” and the evidence for real-world feasibility is weak on multiple levels. Their oracle retrieval guarantees at least one relevant episode plus a small number of distractors (3–7 retrieved items in total) that are directly prepended to the context; in the RL setting, the retrieved information is even provided as “cached memory states.” As a result, the takeaway is closer to “if you hand the model the right evidence and put it in-context, performance improves,” rather than “the system remains reliable under a large-scale memory bank and noisy/approximate retrieval.”

**Requested Changes:**

1. Replace or complement oracle retrieval with at least one realistic retrieval mechanism (e.g., learned retriever over an episode store, nearest-neighbor baselines, or standard RAG-style retrievers) and report performance under non-oracle retrieval. The current evidence is primarily an oracle upper bound and does not establish feasibility in realistic settings.
2. Add robustness/sensitivity analyses for retrieval: vary retrieval quality (probability of including a relevant episode), number of distractors, memory size, and context budget; report degradation curves. This is necessary to support claims about retrieval as a practical complement to parametric learning.
3. Strengthen baseline comparisons against training-time alternatives that directly target the same latent tests (e.g., closure/augmentation objectives for reversals/multi-hop; replay/preplay-style amortization), with compute and data matched as closely as possible. This is needed to justify the claimed “unique” role of episodic retrieval.
4. Clarify and/or narrow claims that imply a broad limitation of gradient-based parametric learning “in present architectures.” At minimum, restate the main result as: oracle episodic reinstatement enables latent-test generalization in these controlled settings, and clearly separate this from any stronger, more general conclusions.

---

> ### Author Response · Authors · 2026-02-11
>
> Thank you for your comments.
>
> Our goal in this paper is to provide insight into where models fail to generalize, and to provide a useful abstraction (the idea of latent learning) for understanding a broad class of these failures, as well as to point towards a class of solutions. We see this as following in a broad line of work that identifies particular generalization challenges for machine learning systems and provide insight into their origins without specifying a full solution to the class of problems (e.g. https://proceedings.mlr.press/v232/de-silva23a.html or https://openreview.net/forum?id=E8m8oySvPJ in this journal). We do point to episodic retrieval as playing a potential role in resolving these issues, but we try to be explicit in the paper that our goal with these oracle demonstrations is to motivate research in this area rather than claiming that we solve the problem ourselves, e.g. in the contributions "Confirming empirically that oracle retrieval can help to overcome this gap, *thus motivating episodic memory systems as an important area of research* for AI." We believe that identifying this class of failures, as well as a direction of investigation for addressing them, could clearly be of interest to researchers in the field.
>
> We use controlled experimental setups in order to have a clear understanding of the types of generalization involved in our tasks; it allows us to ensure that models learn the key knowledge and in-context reasoning abilities, while knowing that our test set is truly held out. We believe that the fact that some of the failures (e.g. the reversal curse) have been observed previously in scaled-up models supports our claim that these are a broader phenomenon that occur beyond toy settings. However, by validating these phenomena in more controlled settings—spanning from language modeling to BC and RL—we hope to provide clearer insight into their origins and potential directions of research that may address them.
>
> Regarding train-time alternatives: we agree that these can resolve some of these problem as well. We explicitly address this in the discussion section on multiplicity of solutions: "We want to highlight that test-time (online) episodic retrieval is not the only solution to the types of problems that we explore. For example, several works have explored training-time (offline) augmentation to address challenges like reversals or multi-hop inferences [...]." As we highlight there, there are advantages and disadvantages to training vs. test-time approaches, and there is some evidence that natural intelligence applies both online and offline replay for potentially similar reasons.
>
> We have revised the abstract (as well as parts of the intro and the "present architectures" sentence to highlight the points above, including clearly stating in the abstract that our experiments use controlled, synthetic settings and are intended to motivate research into episodic retrieval systems rather than to demonstrate a working solution. We hope this will address your concerns about the claims, but we are happy to revise further if there are remaining points of concern.

---

### Review · Reviewer_3Tro · 2026-03-09

**Summary Of Contributions:**

This paper identifies several latent learning challenges where current systems relying purely on parametric learning tend to fail. The authors introduce the concept of latent learning tasks within a clear and rigorous formal framework, carefully distinguishing them from other forms of generalization commonly studied in machine learning.

To support their claims, the paper presents a series of experiments across multiple benchmarks, providing compelling empirical evidence that contemporary parametric models struggle to solve these latent learning tasks. These results highlight important limitations of current approaches that depend exclusively on internalized knowledge.

The authors further demonstrate that a simple oracle retrieval mechanism can successfully address these challenges. This finding offers useful insight into how retrieval can support certain forms of generalization that parametric models alone fail to achieve. In doing so, the work provides an intuitive explanation for why retrieval-augmented generation (RAG) approaches may be beneficial in overcoming some of the limitations of purely parametric systems.

**Audience:**

Yes

**Audience Explanation:**

This paper provides through detailed analysis a better understanding of the challenges linked to latent learning and how to potentially solve them using systems based on retrieval.

**Broader Impact Concerns:**

I do not have concerns about ethical application as the paper only describe limitation of some current form of machine learning.

**Claims And Evidence:**

Yes

**Claims Explanation:**

The claims made in the paper are supported by a series of well-explained experiments conducted on five diverse benchmarks, each contributing in a different way to substantiate the central argument. The results are clearly analyzed, and from a statistical perspective the data appear to be handled appropriately, with error bars reported where necessary.

**Requested Changes:**

Figure 3: When the error bar is zero, the plot does not display a visible marker indicating that an error bar exists. It would be helpful to make this more visible (e.g., by ensuring that a point marker is still shown).

Figure 5: For most of the plots, the statistical significance of the results appears sufficiently clear, making detailed reporting less critical. However, it would still be useful to compute and report statistical significance for the results shown in Figure 5a (for example, for the syllogism task).

Section 3: This section introduces terminology related to latent learning tasks (e.g., “task cue”), but this terminology is not reused when describing the different benchmarks later in the paper. Reframing the benchmark descriptions using the terminology introduced earlier would likely improve clarity and consistency.

---

> ### Author Response · Authors · 2026-04-05
>
> We thank the reviewer for their thoughtful and supportive review! We have incorporated the terminology in the benchmark descriptions (noting in a few places where it is limited from capturing the full subtlety of the issues, as was already noted in Section 3), as well as the statistical tests. For the errorbars, we found it difficult to make the plotting library produce this in a way that wasn't misleading. However, we have added a note to the caption of this and the other figures where it appears explicitly clarifying the point.

---

### Review · Reviewer_PUqD · 2026-04-14

**Summary Of Contributions:**

The paper frames latent learning, acquiring information not relevant to the current training task but useful for future ones, as a key gap between natural and artificial intelligence. Through four synthetic benchmarks (codebooks, simple reversals, semantic structure, gridworld navigation), the authors show that standard transformer training fails on test conditions requiring latent generalization, despite the model possessing all necessary knowledge components individually. An oracle episodic retrieval system largely resolves these failures by reinstating relevant past episodes into context. A secondary finding is that within-episode ICL training sequences are necessary for retrieval benefits to materialize. The work is motivated by complementary learning systems theory and draws an analogy to hippocampal contributions to flexible generalization.

Strengths: The experimental design is clean and the ablations are well-controlled. The ICL-retrieval interaction finding is the most novel empirical contribution.

Weaknesses: All experiments use small transformers trained from scratch on synthetic data, which limits how much can be inferred about the large pretrained model phenomena that motivate the work, and the oracle retrieval assumption leaves the practical gap to any realistic retrieval system uncharacterized.

**Audience:**

Yes

**Audience Explanation:**

The latent learning framing offers a unifying perspective on several actively studied failure modes of language models, including the reversal curse and multi-hop failures. The connection to complementary learning systems theory will also be of interest to people working at the intersection of machine learning and cognitive science.

**Claims And Evidence:**

Yes

**Claims Explanation:**

The core negative claim, that parametric learning fails on latent tests while passing all prerequisite component tests, is cleanly demonstrated across multiple settings, and the ablations rule out batch size and data augmentation as confounders.

The positive claim rests entirely on oracle retrieval, which largely confirms that in-context reasoning works when the right document is provided, a fact already established by the reversal-with-context baselines in the same figures. The distance between this and any realistic retrieval system is large and uncharacterized. The semantic structure results are also less clear, with weak and inconsistent retrieval benefits.

**Requested Changes:**

The oracle retrieval assumption leaves the practical gap to any realistic retrieval system entirely uncharacterized. How sensitive are the observed gains likely to be to retrieval precision, even qualitatively?

The weak and inconsistent retrieval benefits on the semantic structure benchmark are attributed to insufficient ICL training examples, but this explanation is post-hoc. Can the authors provide an experiment on that benchmark that isolates this factor?

Can the authors discuss how the findings are expected to extend (or not) to large pretrained models?

---

> ### Author Response · Authors · 2026-04-28
>
> Thanks to the reviewer for their thoughtful comments; we have edited the paper to try to address the concerns and questions, and we outline these changes below.
>
> > The oracle retrieval assumption leaves the practical gap to any realistic retrieval system entirely uncharacterized. How sensitive are the observed gains likely to be to retrieval precision, even qualitatively?
>
> In our experiments, recall was perfect but precision was not all that high — most the retrieved epsiodes in any given instance were not relevant to the task at hand (e.g. 1/8 = 12.5% precision), yet we still see high accuracy in our retrieval experiments. Two caveats: 1) Of course, the results are much more sensitive to recall; clearly the model must revert to parametric knowledge when no relevant information is recalled into context. 2) Precision may need to be higher during learning than at test time; indeed, this is one of the key benefits of the necessity-of-ICL-sequences experiments (which effectively introduce higher-precision sequences, since the relevant information appears clearly in the present-document context with relatively few distractors there). We have added some discussion of this to the paper in a new limitations section. We also have attempted to launch a sweep where we further decrease retrieval precision to evaluate its effects on the results, but due to cluster contention (in the run up to the NeurIPS deadline) we haven’t been able to complete most experiments before the response deadline. We hope to add the results to our next revision, though.
>
> > The weak and inconsistent retrieval benefits on the semantic structure benchmark are attributed to insufficient ICL training examples, but this explanation is post-hoc. Can the authors provide an experiment on that benchmark that isolates this factor?
>
> We indeed thought about running an experiment where we enhanced ICL training to test this hypothesis, but unfortunately it is not easy to introduce more ICL sequences into the benchmark (the code makes strong assumptions about the type of structure the data are generated from, and constructing ICL examples from that structure that have sufficient diversity for illustrating the different types of relations at play, but that do not contaminate the test set, is hard without a complete redesign of the benchmark). We apologize for not being able to test this more directly; we have instead revised this statement in both places it occurs to emphasize its status as a possible hypothesis for explaining the findings, and not a definitive claim.
>
> > Can the authors discuss how the findings are expected to extend (or not) to large pretrained models?
>
> We already touched on some related observations in large pretrained models in the related work, but we have expanded this further in a new paragraph at the beginning of the discussion.  We have also added some discussion of the possible differences (and the tradeoffs of our small-scale experiments) to the limitations section.

---

### Decision · Action_Editor_8D6A · 2026-05-30

**Recommendation:** Accept as is

**Additional Comments:**

I still encourage the authors to address and incorporate comments from the reviewers. I do not expect any big changes, particularly those requested by H96m, but their perspectives can help make the paper and framing better.

**Audience:**

Yes

**Audience Explanation:**

This paper gives insights into how models fail to generalize, and provides some evidence that episodic memory can complement parametric learning to provide better generalization. As highlighted by the reviewers, the experimental setting does not match typical (big) models used in practice, but the existence of the phenomena helps understand potential behavior in the wild. Such a foundation can help other researchers build on these insights.

**Claims And Evidence:**

Yes

**Claims Explanation:**

The authors carefully designed their experiments to make clear claims, using synthetic settings and smaller models. The results are thorough and clear, and the motivation well explained.